# PRECISION HIGHWAY
# FOR ULTRA LOW-PRECISION QUANTIZATION

## ABSTRACT

Neural network quantization has an inherent problem called *accumulated quantization error*, which is the key obstacle towards ultra-low precision, e.g., 2- or 3-bit precision. To resolve this problem, we propose *precision highway*, which forms an *end-to-end* high-precision information flow while performing the ultra-low-precision computation. First, we describe how the precision highway reduce the accumulated quantization error in both convolutional and recurrent neural networks. We also provide the quantitative analysis of the benefit of precision highway and evaluate the overhead on the state-of-the-art hardware accelerator. In the experiments, our proposed method outperforms the best existing quantization methods while offering 3-bit weight/activation quantization with no accuracy loss and 2-bit quantization with a 2.45 % top-1 accuracy loss in ResNet-50. We also report that the proposed method significantly outperforms the existing method in the 2-bit quantization of an LSTM for language modeling.

## 1 INTRODUCTION

Energy-efficient inference of neural networks is becoming increasingly important in both servers and mobile devices (e.g., smartphones, AR/VR devices, and drones). Recently, there have been active studies on ultra-low-precision inference using 1 to 4 bits (Rastegari et al., 2016; Hubara et al., 2016; Zhu et al., 2016; Zhou et al., 2016; 2017; Zhuang et al., 2018; Choi et al., 2018b; Liu et al., 2018; Choi et al., 2018a) and their implementations on CPU and GPU (Tulloch & Jia, 2017), and dedicated hardware (Park et al., 2018; Sharma et al., 2017). However, as will be explained in section 5.2, the existing quantization methods suffer from a problem called *accumulated quantization error* where large quantization errors get accumulated across layers, making it difficult to enable ultra-low precision in deep neural networks.

In order to address this problem, we propose a novel concept called *precision highway* where an end-to-end path of high-precision information reduces the accumulated quantization error thereby enabling ultra-low-precision computation. Our proposed work is similar to recent studies (Liu et al., 2018; Choi et al., 2018b) which propose utilizing pre-activation residual networks, where skip connections are kept in full precision while the residual path performs low-precision computation. Compared with these works, our proposed method offers a generalized concept of high-precision information flow, namely, precision highway, which can be applied to not only the pre-activation convolutional networks but also both the post-activation convolutional and recurrent neural networks. Our contributions are as follows.

- We propose a novel idea of network-level approach to quantization, called *precision highway* and quantitatively analyze its benefits in terms of the propagation of quantization errors and the difficulty of convergence in training based on the shape of loss surface.

- We provide the detailed analysis of the energy and memory overhead of precision highway based on the state-of-the-art hardware accelerator model. According to our experiments, the overhead is negligible while offering significant improvements in accuracy.

- We apply precision highway to both convolution and recurrent networks. We report a 3-bit quantization of ResNet-50 without accuracy loss and a 2-bit quantization with a very small accuracy loss. We also provide the sub 4-bit quantization results of long short-term memory (LSTM) for language modeling.

## 2 RELATED WORK

Migacz (2017) presented an int8 quantization method that selects an activation truncation threshold to minimize the Kullback-Leibler divergence between the distributions of the original and quantized data. Jacob et al. (2017) proposed a quantization scheme that enables integer-arithmetic only matrix multiplications (practically, 8-bit quantization for neural networks). These methods are implemented on existing CPUs or GPUs (Jacob et al., 2015; ACL).

Hubara et al. (2016) presented a binarization method and demonstrated the performance benefit on a GPU. Rastegari et al. (2016) proposed a binary network called XNOR-Net in which a weight-binarized AlexNet gives the same accuracy as a full-precision one. Zhou et al. (2016) presented DoReFa-Net, which applies $tanh$-based weight quantization and bounded activation. Zhou et al. (2017) proposed a balanced quantization that attempts to balance the population of values on quantization levels. He et al. (2016b) proposed utilizing full precision for internal cell states in the LSTM because of their wide value distributions. This work is similar to ours in that high-precision data are selectively utilized to improve the quantized network. Our difference is proposing a network-level end-to-end flow of high-precision activation. Recently, Zhuang et al. (2018) presented 4-bit quantization with ResNet-50. They adopt Dorefa-net style weight quantization with static bounded activation, and improve accuracy by adopting multi-step quantization and knowledge distillation during fine-tuning. Mishra et al. (2017) proposed a trade-off between the number of channels and precision. Clustering-based methods have the potential to further reduce the precision (Han et al., 2015). However, they require a lookup table and full-precision computation, which makes them less hardware-friendly.

Recently, Choi et al. (2018b;a) and Liu et al. (2018) proposed utilizing full-precision on the skip connections in pre-activation residual networks[1]. Compared with those works, our proposed idea has a salient difference in that it offers a network-level solution and demonstrates that the end-to-end flow of high-precision information is crucial. In addition, our method is not limited to pre-activation residual networks, but general enough to be applied to both post-activation convolutional and recurrent neural networks.

## 3 PROPOSED METHOD

In *precision highway*, we build a path from the input to output of a network to enable the end-to-end flow of high-precision activation, while performing low-precision computation. Our proposed method was motivated (1) by a residual network where the signal, i.e., the activation/gradient in a forward/backward pass, can be directly propagated from one block to another (He et al., 2016a) and (2) by the LSTM, which provides an uninterrupted gradient flow across time steps via the inter-cell state path (Gers et al., 2000). Our proposed method focuses instead on improving the accuracy of quantized network by providing an end-to-end high-precision information flow.

In this section, we first describe the precision highway in the cases of residual network (section 3.1) and recurrent neural network (section 3.2). Then, we discuss practical issues to be addressed before application to other networks in section 3.3.

### 3.1 PRECISION HIGHWAY ON RESIDUAL NETWORK

In the case of a residual network, we can form a precision highway by making high-precision skip connections. In this subsection, we explain how high-precision skip connections can be constructed to reduce the accumulated quantization error.

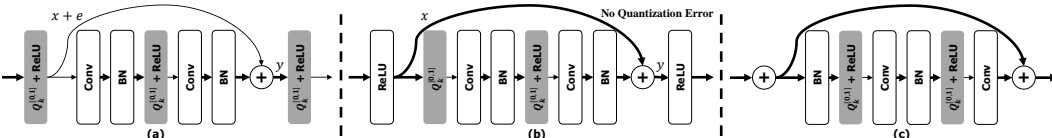

Figure 1: Comparison of conventional quantization and our proposed idea on residual network.

---

[1] This work was conducted in parallel with Choi et al. (2018b;a); Liu et al. (2018)

In the conventional residual block shown in Figure 1 (a), quantization (denoted as $Q_k^{[0,1]}$, k-bit linear quantization in range from 0 to 1) is applied to all of the activations after the activation function. In the figure, thick (thin) arrows represent high-precision (low-precision) activations. As the figure shows, the input of a residual block is first quantized, and the quantized input ($x + e$ in the figure), which contains the quantization error $e$, enters both the skip connection and residual path. The output of a residual block, $y$, is calculated as follows:

$$y = F(x + e) + x + e = F(x) + x + e_r + e, \tag{1}$$

where $F()$ represents a residual function (typically, 2 or 3 consecutive convolutional layers). For simplicity of explanation, we assume that $F(x + e)$ can be decomposed into $F(x) + e_r$, where $e_r$ represents the resulting quantization error of the residual path incurred by the quantization operations on the residual path as well as the quantization error in the input, $e$. As the equation shows, output $y$ has two quantization error terms, that of residual path, $e_r$, and that of the skip connection, $e$.

Figure 1 (b) shows our idea of high-precision skip connection. Compared with Figure 1 (a), the difference is the location of the first quantization operation in the residual block. In Figure 1 (b), quantization is applied only to the residual path after the bifurcation to the residual path and skip connection. As shown in the figure, the skip connection now becomes a thick arrow, i.e., a high-precision path. The proposed idea gives the output of the residual block as follows:

$$y = F(x + e) + x = F(x) + x + e_r. \tag{2}$$

As Equation 2 shows, the proposed idea eliminates the quantization error of skip connection $e$. Thus, only the quantization error of the residual path $e_r$ remains in the output of the residual block. Note that all of the input activations of the residual path are kept in low precision. It enables us to perform low-precision convolution operations in the residual path. We keep high-precision activation only on the skip connection and utilize it only for the element-wise addition. As will be shown in our experiments, the overhead of computation and memory access cost is small since the element-wise addition is much less expensive than the convolution on the residual path, and the low-precision activation is accessed for the computation on the residual path.

As will be shown later, our method gives a smaller quantization error, and the gap between the quantization error of the existing method and that of ours becomes wider across layers. Because of the reduction of the accumulated quantization error, the proposed method offers much better accuracy than the state-of-the-art methods with an ultra-low precision of 2 and 3 bits.

Note also that, as shown in figure 1 (c), our idea can be applied to other types of residual blocks, including the full pre-activation residual block (He et al., 2016a) as proposed in some recent works (Choi et al., 2018a; Liu et al., 2018). However, our idea is general in that it is applicable to recurrent networks as well as post-activation convolutional networks. Especially, our proposed idea is advantageous over the existing ones since hardware accelerators tend to be designed assuming as the input non-negative input activations enabled by ReLU activation functions (Park et al., 2018; Kim et al., 2018). Contrary to the existing works (Choi et al., 2018a; Liu et al., 2018), we provide a detailed analysis of the effect of precision highway.

## 3.2 Precision Highway on Recurrent Neural Network

Figure 2 illustrates how the precision highway can be constructed on the LSTM (Gers et al., 2000). In time step $t$, the LSTM cell takes, as an input, new input $x_t$, along with the results of the previous time step, output $h_{t-1}$ and cell state $c_{t-1}$. First, it calculates four intermediate signals: $i$ (input gate), $f$ (forget gate), $g$ (gate gate), and $o$ (output gate). Then, it produces two results, $c_t$ and $h_t$, as follows:

$$i_t = \sigma(W_{ii}x_t + b_{ii} + W_{hi}h_{(t-1)} + b_{hi}), \tag{3a}$$

$$f_t = \sigma(W_{if}x_t + b_{if} + W_{hf}h_{(t-1)} + b_{hf}), \tag{3b}$$

$$g_t = \tanh(W_{ig}x_t + b_{ig} + W_{hg}h_{(t-1)} + b_{hg}), \tag{3c}$$

$$o_t = \sigma(W_{io}x_t + b_{io} + W_{ho}h_{(t-1)} + b_{ho}), \tag{3d}$$

$$c_t = f_t \odot c_{(t-1)} + i_t \odot g_t, \tag{3e}$$

$$h_t = o_t \odot \tanh(c_t), \tag{3f}$$

where $\sigma$ represents a sigmoid function, $\odot$ the element-wise multiplication, $W$ the weight matrix, and $b$ the bias.

Figure 2: Comparison on residual network.

In the conventional LSTM operation, as Figure 2 (a) shows, the quantization (gray box denoted by $Q_k$ with the output value range as the superscript) is applied to all of the activations before computation. The results of a time step, $c_t$ and $h_t$, are calculated based on such inputs with quantization errors. More specifically, cell state $c_t$ is calculated with the quantized, i.e., low-precision, inputs of $c_{t-1}$, $f$, $i$, and $g$. Thus, cell state $c_t$ accumulates the quantization errors of those inputs. In addition, output $h_t$ also accumulates the quantization errors from its inputs, $c_t$ and $o$. Then, they are propagated to the next time steps. Thus, we have the problem of accumulated quantization error across the time steps. Such an accumulation of quantization error will prevent us from achieving ultra-low precision.

Figure 2 (b) shows how we can build the precision highway in the LSTM cell. The figure shows that the quantization operation is applied only to the inputs of matrix multiplication (a circle denoted with $\times$ in the figure). Thus, all of the other operations and their input activations are in high precision. Specifically, when calculating $c_t$, the inputs are not quantized, which reduces the accumulation of quantization error on $c_t$. The computation of $h_t$ can also reduce the accumulation of quantization error by utilizing high-precision inputs. The construction of such a precision highway allows us to propagate high-precision information, i.e., cell states $c_t$ and outputs $h_t$, across time steps.

Note that we benefit from low-precision computation by performing low-precision matrix multiplications (in Equations 3a-3d), which dominate the total computation cost. In our proposed method, all of the element-wise multiplications in Equations 3e and 3f are performed in high precision. However, the overhead of this high-precision element-wise multiplications is negligible compared with the matrix multiplication in Equations 3a-3d. In addition, this method can be applied to other types of recurrent neural networks. For instance, the GRU (Chung et al., 2014) can be equipped with a precision highway, in a way similar to that shown in Figure 2 (b), by keeping high-precision output $h_t$ while performing low-precision matrix multiplications and high-precision element-wise multiplications.

### 3.3 PRACTICAL ISSUES WITH PRECISION HIGHWAY

In order to generalize our proposed idea to other networks in real applications, we need to address the following issues. First, in the case of feed-forward networks with identity path, our precision highway idea is applicable regardless of pre-activation or post-activation structure. We can exploit the benefit of reduced precision by applying quantization in front of matrix multiplications, while maintain the accuracy by handing the identity path in high precision. Second, in the case of non-residual feed-forward networks, the precision highway can be constructed by equipping them with additional skip connections. In the case of networks with multiple candidates for the precision highway, e.g., DenseNet, which has multiple parallel skip connections (Huang et al., 2017), we need to address a new problem of selecting skip connections to form a precision highway, which is left for future work.

## 4 TRAINING

In this section, we describe weight quantization and fine-tuning for weight/activation quantization.

### 4.1 LINEAR WEIGHT QUANTIZATION BASED ON LAPLACE DISTRIBUTION MODEL

Figure 3 illustrates that a Laplace distribution can well fit the distributions of weights in full-precision trained networks. Thus, we propose modeling the weight distribution with Laplace distribution and selecting quantization levels for weights based on a Laplace distribution.

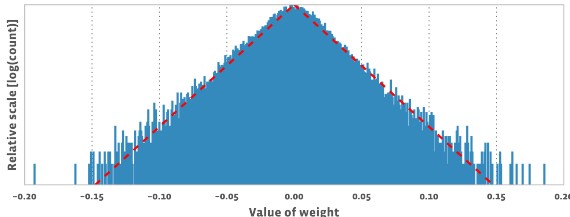 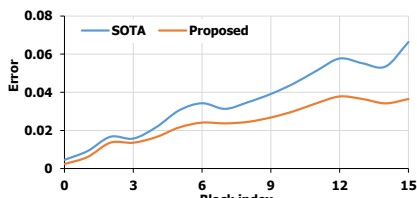

Figure 3: Weight histogram and Laplace approximation (dashed line) of the convolutional layer of a trained full-precision ResNet-50.

Figure 4: Quantization error accumulation across residual blocks in ResNet-50.

Given a distribution of weights and a target precision of $k$ bits, e.g., 2 bits, the quantization levels are determined as follows. First, the quantization levels for $k$ bits are pre-computed for the normalized Laplace distribution. We determine quantization levels that minimize L2 error on the normalized Laplace distribution. For instance, in case of the 2-bit quantization, the error is minimized when four quantization levels are placed evenly with a spacing of 1.53 $\mu$, where $\mu$ is the mean of the absolute value of weights. Given the distribution of weights and the pre-calculated quantization levels on the normalized Laplace distribution for the given $k$ bits, we determine the real quantization levels by multiplying the pre-computed quantization levels and the mean of the absolute value of weights.

Our proposed weight quantization is similar to the one in Choi et al. (2018a). Compared to it, ours is simpler in that only Laplace distribution model is utilized, and our experiments show that the precision highway together with the proposed simple weight quantization gives outstanding results.

## 4.2 FINE-TUNING FOR WEIGHT/ACTIVATION QUANTIZATION

Our quantization is applied during fine-tuning after training a full-precision network. As the baseline, we adopt the fine-tuning procedure in (Zhuang et al., 2018), where we perform incremental/progressive quantization. In contrast to (Zhuang et al., 2018), we first quantize activations and then weights in an incremental quantization. In addition, for each precision configuration, we perform teacher-student training to improve the quantized network (Zhuang et al., 2018; Mishra & Marr, 2017). As the teacher network, we utilize a deeper full-precision network, e.g., ResNet-101, compared to the student network, e.g., quantized ResNet-50. Note that, during fine-tuning, we apply quantization in forward pass while updating full-precision weights during backward pass.

## 5 EXPERIMENTS

### 5.1 EXPERIMENTAL SETUP

We implemented the proposed method in PyTorch and Caffe2. We use two types of trained neural networks, ResNet-18/50 for ImageNet and an LSTM for language modeling (Zaremba et al., 2014; Press & Wolf, 2016; Inan et al., 2016). We evaluate 4-, 3-, and 2-bit quantizations for the networks.

For ResNet, we did test with single center crop of 256x256 resized image. We compare our proposed method with the state-of-the-art methods (Zhou et al., 2016; Zhuang et al., 2018; Choi et al., 2018b;a; Liu et al., 2018). Note that, for the teacher-student training, we use the same teacher network for both the baseline method (our implementation) (Zhuang et al., 2018) and ours. We also evaluate the effects of increasing the number of channels (Mishra et al., 2017) to recover from accuracy loss due to quantization. As in the previous works (Hubara et al., 2016; Zhou et al., 2017; Zhuang et al., 2018; Choi et al., 2018b;a; Liu et al., 2018; Mishra et al., 2017), we do not apply quantization to the first and last layers.

The LSTM has 2 layers and 300 cells on each layer. We used the Penn Treebank dataset and evaluated the perplexity per word. We compared the state-of-the-art method in (He et al., 2016b) and our proposed method.

## 5.2 Analysis of Accumulated Quantization Error

Figure 4 shows the quantization errors across layers in ResNet-50 when applying the state-of-the-art 4-bit quantization to activations. We prepared, from the same initial condition, two activation-quantized networks (one with precision highway and the other with low precision skip connection) where weights are not modified and only activations are quantized to 4 bits. As the metric of the quantization error, we utilize a metric based on the cosine similarity between the activation tensor of corresponding layer in the full-precision and quantized networks, respectively.

As the figure shows, in the existing method, the quantization errors become larger for deeper layers. It is because the quantization error generated in each layer is propagated and accumulated across layers. We call this *accumulated quantization error*. The accumulated errors become larger with more aggressive quantization, e.g., 2 bits, and cause poor performance, i.e., 4.8 % drop (Zhuang et al., 2018) from the top-1 accuracy of the full-precision ResNet-50 for ImageNet classification.

The accumulation of quantization errors is an inherent characteristic of a quantized network in both feed-forward and feed-back networks. In the case of a recurrent neural network, the quantization errors are propagated across time steps. As shown in Figure 4, our proposed precision highway significantly reduces the accumulated quantization errors, which enables 3-bit quantization without accuracy drop and much better accuracy in 2-bit quantization than the existing methods.

## 5.3 Loss Surface Analysis of Quantized Model Training

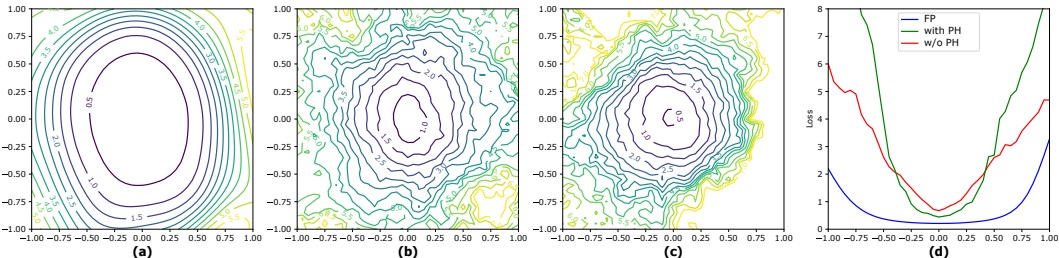

Figure 5: Loss surface of ResNet-18 on Cifar-10: (a) full-precision model (FP), (b) 1-bit activation and 2-bit weight quantized model (1A2W) without precision highway (PH), (c) 1A2W with precision highway, and (d) cross-section of loss surface.

Figure 5 visualizes the complexity of loss surface depending on the existence of precision highway. We obtained the figures by applying the method proposed by Li et al. (Li et al., 2017). Each figure represents loss surface seen from the local minimum we obtained from the training, i.e., the weight vector of the final trained model. The origin of the figure at (0, 0) corresponds to the weight vector of the local minimum. As shown in the figure 5 (d), the precision highway gives better loss surface (having lower and smoother surface near the minimum point and steep and simple surface elsewhere) than the existing quantization method. This characteristic helps stochastic gradient descent (SGD) method to quickly converge to a good local minimum offering better accuracy than the existing method.

## 5.4 Evaluating the Accuracy of Quantized Model

Table 1: 2-bit quantization results. Top-1/Top-5 accuracy [%].

| Laplace | Teacher | Highway | ResNet-18 | ResNet-50 |
|:---:|:---:|:---:|:---:|:---:|
| ✓ | | | 61.66 / 84.28 | 70.50 / 89.84 |
| ✓ | ✓ | | 62.66 / 85.00 | 71.70 / 90.39 |
| ✓ | | ✓ | 65.83 / 86.71 | 72.99 / 91.19 |
| ✓ | ✓ | ✓ | 66.71 / 87.40 | 73.55 / 91.40 |
| **Full-precision** | | | 70.15 / 89.27 | 76.00 / 92.98 |
| **Zhuang's (ours)** | | | 60.06 / 83.34 | 69.04 / 89.14 |
| **Zhuang's (ours) + Teacher** | | | 61.21 / 84.36 | 70.48 / 89.83 |

Table 2: Comparison of accuracy loss in 2-bit activation/weight quantization. Bi-real applies 1-bit activation/weight quantization.

| | ResNet-18 | ResNet-50 |
|:---:|:---:|:---:|
| **Ours** | 3.44 | 2.45 |
| **DoReFa** (Zhou et al., 2016) | 7.6 | 9.8 |
| **Zhuang's** (Zhuang et al., 2018) | - | 4.8 |
| **PACT** (Choi et al., 2018b) | 5.8 | 4.7 |
| **PACT_new** (Choi et al., 2018a) | 3.4 | 2.7 |
| **Bi-Real** (Liu et al., 2018) | 12.9 | - |

Table 1 shows the accuracy of 2-bit quantization for ResNet-18/50. We evaluate each of our proposed methods, Laplace, teacher, and highway, as shown in the table. When the highway box is unchecked the skip connection is branched after the quantization and when the teacher box is unchecked, we use the conventional cross-entropy loss. Compared with the full-precision accuracy, our 2-bit quantization (when all the methods were applied) gives a top-1 accuracy of 73.55 %, which is within 2.45 % of the full-precision accuracy and much better than the state-of-the-art method (Zhuang's 70.8 %) having a top-1 accuracy loss of 4.8 %. Note that Zhuang's implemented all the methods, incremental/progressive quantization and teacher-student training, in Zhuang et al. (2018). We presents the accuracy results of our own implementations of Zhuang's method under the same amount of training time. Zhuang's (ours) implemented only incremental and progressive methods while Zhuang's + Teacher utilized our teacher network.

Table 1 shows the effects of the precision highway. Compared with our solution supporting only Laplace and teacher, the highway provides an additional gain of 1.85 % (71.70 % to 73.55 %) in the top-1 accuracy of ResNet-50. The effects of the Laplace method can be evaluated by comparing the result of our implementation of Zhuang's (69.04 % of top-1 accuracy in ResNet-50) and that of our solution adopting the Laplace model (70.50 %) because these are the same except for the weight quantization method, i.e., $tanh$ vs. Laplace based model. The Laplace method gives 1.46 % better accuracy. The table indicates that ResNet-18 also benefits from our proposed methods like ResNet-50.

Table 2 compares the additional accuracy loss of quantization methods with respect to full-precision accuracy. The table shows that ours significantly outperform the methods without precision-highway (DoReFa, Zhuang's, and PACT). PACT_new and Bi-Real utilize high-precision skip connections on pre-activation resiudal networks. Thus, they show comparable results to ours[2]. Note that our results in the table are obtained from the conventional post-activation residual network, which demonstrates the generality of our proposed precision highway. As will be shown below for the LSTM, our proposed method is generally applied to recurrent networks as well as feed forward ones.

Table 3: Impact of highway precision (y-axis: low precision and x-axis: highway precision). Top-1/Top-5 accuracy [%].

| | | Full | 8-bit | 6-bit | | | | Full | 8-bit | 6-bit |
|---|---|---|---|---|---|---|---|---|---|---|
| | 4-bit | 71.05 / 90.16 | 71.07 / 90.20 | 70.54 / 89.77 | | | 4-bit | 76.92 / 93.44 | 76.69 / 93.27 | 76.25 / 93.13 |
| **ResNet-18** | 3-bit | 70.29 / 89.54 | 70.08 / 89.51 | 69.39 / 88.95 | | **ResNet-50** | 3-bit | 76.20 / 93.09 | **76.08** / 93.03 | 75.33 / 92.63 |
| | 2-bit | 66.71 / 87.40 | 66.62 / 87.33 | 65.26 / 86.47 | | | 2-bit | 73.55 / 91.40 | 73.15 / 91.34 | 72.79 / 91.20 |

Table 3 shows the impact of the precision of the precision highway. We obtained the results by varying the highway precision (without retraining) after obtaining the results with the full-precision highway. The table shows that 2-bit quantization with the 8-bit highway gives only 0.09 % and 0.40 % drops in the top-1 accuracy for ResNet-18 and ResNet-50, respectively, from that of the 2-bit quantization with the full-precision highway. Most importantly, our 3-bit quantization (with the 8-bit highway) gives the same accuracy as the full-precision network, i.e., 76.08 % in ResNet-50, which means that our proposed method reduces the precision of the ResNet-50 from 4 bits with Zhuang et al. (2018) down to 3 bits even with the 8-bit highway.

Table 4: Accuracy of wide ResNet-18 and ResNet-50 with quantization. Top-1/Top-5 accuracy [%].

Table 5: Perplexity of quantized LSTM. (x,y) means x-bit weight/y-bit activation.

| | Full | 3-bit | 2-bit | Zhuang's (ours) 2-bit | | | (4,4) | (3,4) | (3,3) | (2,3) | (2,2) |
|---|---|---|---|---|---|---|---|---|---|---|---|
| **wResNet-18** | 74.60 / 91.85 | 75.49 / 92.45 | **73.80** / 91.56 | 70.81 / 90.02 | | **Without Highway** | 97.21 | 98.14 | 105.33 | 107.45 | 133.25 |
| **wResNet-50** | 77.78 / 93.87 | 78.45 / 94.28 | **77.35** / 93.69 | 75.54 / 92.71 | | **With Highway** | 95.94 | 96.29 | 100.77 | 102.55 | 114.44 |

Table 4 shows the effects of two times wider channel under 2-bit quantization. We first doubled the number of channels in ResNet-18 and ResNet-50, and then quantized them with our methods. As the table shows, the wide ResNets give better accuracy than the full-precision ones even for 2-bit quantization, i.e., 73.80 % (77.35 %) in Table 4 vs. 70.15 % (76.00 %) of the full precision in Table 1

---

[2]We performed 1-bit activation/weight quantization for the post-act style ResNet-18. For a fair comparison, we didnt apply the teacher-student and progressive quantization method and instead adopted BN-retraining proposed in Bi-Real Net. Our 1-bit activation/weight ResNet-18 gives 56.73 / 80.11 % of Top-1/Top-5 accuracy, which is by 0.33 / 0.61 % higher than the result of Bi-Real Net, respectively.

for ResNet-18 (ResNet-50). It would be worth investigating how to minimize the channel size while meeting the full-precision accuracy with ultra-low precision, which is left for future work.

Table 5 lists the quantization results for the LSTM. We varied the bit configuration (weight, activation) and obtained the perplexity results (the lower, the better). The table shows that our proposed method significantly reduces the perplexity. Compared with the perplexity of full-precision model (92.84), our 4-bit quantization gives a very small increase of 3.3 % (92.84 to 95.94). The precision highway provides more gain for a more aggressive quantization. Specifically, it reduces the perplexity by 14.1 % (from 133.25 to 114.44) in the 2-bit quantization, (2,2). Compared with the state-of-the-art quantization of a similar LSTM (He et al., 2016b)[3], ours offers much better results, i.e., a much smaller increase in perplexity, e.g., a 23.3 % increase (92.84 to 114.44 in Table 5) vs. a 39.4 % increase (109 to 152 in (He et al., 2016b)) in perplexity for 2-bit quantization.

## 5.5 HARDWARE COST EVALUATION OF QUANTIZED MODEL

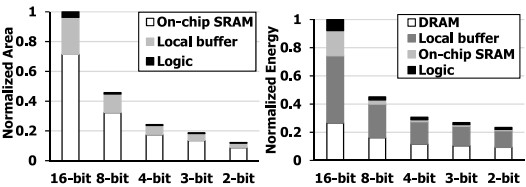

Figure 6: Comparison of chip area and energy consumption on the hardware accelerator.

Table 6: Number of operations. * denotes the high-precision operation.

|  | LSTM (300) | ResNet-18 | ResNet-50 |
|---|---|---|---|
| Low-precision MAC | 720 K | 6.89 G | 15.1 G |
| High-precision Add | 0.3 K | 9.68 M | 62.0 M |
| Non-linear Op* | 1.5 K | 7.48 M | 39.9 M |
| Elt-wise Multi* | 0.9 K | - | - |

Figure 6 shows the chip area cost and energy consumption of ResNet-18 at different levels of precision on the state-of-the-art hardware accelerator (Chen et al., 2017). The accelerator is synthesized at 65 nm, 250 MHz, and 1.0 V. Each processing element (PE) consists of a multiply-accumulate (MAC) unit and local buffers. The PEs share global on-chip 2 MB static random access memory (SRAM) at 16-bit precision and the size of which is adjusted proportional to the precision. As the figure shows, the reduced precision offers significant reduction in chip area, e.g., 82.3 % reduction from 16 to 3 bits and energy consumption, e.g., 73.1 % from 16 to 3 bits. In the 2-bit case where the overhead of precision highway is the largest, the precision highway incurs only 3.9 % additional energy consumption due to the high-precision data while offering 4.1 % better accuracy than the case that precision highway is not adopted. The accelerator is already equipped large internal buffer for partial sum accumulation. Thus, precision highway incurs additional energy consumption mainly on the accesses to on-chip SRAM and main memory (dynamic random access memory, DRAM).

Table 6 compares the number of operations in three neural networks used in our experiments. The table explains why the high-precision operations incur such a small overhead in energy consumption. As the table shows, it is because the frequency of high-precision operations is much smaller than that of low-precision operations. For instance, the 2-bit LSTM network has one high-precision (in 32 bits) element-wise multiplication for every 800 2-bit multiplications.

## 6 CONCLUSION

In this paper, we proposed the concept of end-to-end precision highway which can be applied to both feedforward and feedback networks and enable ultra-low precision in deep neural networks. The proposed precision highway reduces quantization errors by keeping high-precision activation from the input to output of the network with small computation costs. We described how it reduces the accumulated quantization error and presented quantitative analyses in terms of accuracy and hardware cost as well as training characteristics. Our experiments showed that the proposed method outperforms the state-of-the-art methods in the 3- and 2-bit quantizations of ResNet-18/50 and 2-bit quantization of an LSTM model. We believe that our work will serve as a step toward mixed precision networks for computational efficiency.

---

[3] Note that we compared their relative change from the full-precision perplexity because the full-precision perplexity of the state-of-the-art method (109) is different from that of ours (92.84).

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
