# OpenReview forum: "Precision Highway for Ultra Low-precision Quantization"
_ICLR.cc/2019/Conference_

### Official Review · AnonReviewer1 · 2018-11-02
**This paper proposes to keep a high activation/gradient flow in two special kinds of networks structures, namely ResNet and LSTM. For ResNet, the skip connections are made high-precision by adding the skip connection before quantization. For LSTM, the cell and hidden state computations are of high precision.**

**Rating:** 5
**Confidence:** 3

**Review:**

The proposed method is advantageous in that it only requires changes to some parts of the original ResNet or LSTM, without having to significantly change the network structure or training algorithm. It also reports empirical success of using high-precision skip connections in ResNet and cell/hidden state updates in LSTMs.

However, it is unclear why it is necessary to keep a high-precision activation/gradient flow. What is the problem with existing quantized networks that do not have these high-precision-flow? Also, how does the high-precision flow interact with the rest of the network (with low-precision operations)?

Moreover, the proposed method has limited novelty as the use of full-precision skip connections has been proposed in Bi-Real (Liu et al. 2018).

Minor:
- It is hard to tell that the weight histogram in Figure 3 is similar to a Laplacian distribution. It can also be approximated by other distributions (such as Gaussian or piecewise-linear distributions).
- What kind of activation quantization is used?
- In the experiments, when is the cosine similarity between the quantized and full-precision networks computed? after training or on an intermediate training step?
- What are the axes in Figure 5? Why is there only one local minimum in Figure 5(d)? Why the training with PH converges even slower than without PH at the early stage of training?

---

> ### Author Response · Authors · 2018-11-19
> **The response for Reviewer 1**
>
>
> 1. The  importance of precision highway
>
> Precision highway helps reduce the accumulated quantization error. In ResNet, the difference between Equations (1) and (2) explains how the precision highway reduces quantization error. Without precision highway, the output of residual block has additional quantization error, ‘e’ in Equation (1) while the precision highway removes it as shown in Equation (2).
> Section 3.2 describes how the precision highway reduces the accumulation of quantization error in the LSTM as follows.
> “Specifically, when calculating ct, the inputs are not quantized, which reduces the accumulation of quantization error on ct. The computation of ht can also reduce the accumulation of quantization error by utilizing high-precision inputs. The construction of such a precision highway allows us to propagate high-precision information, i.e., cell states ct and outputs ht, across time steps”
> The result of low-precision computation is in high precision before quantization. We perform elementwise operations (additions in ResNet and multiplications in LSTM/GRU) between the precision highway and the high-precision result of low-precision computation. In other words, the elementwise computations in ResNet and LSTM/GRU are performed in high precision as mentioned in the original manuscript as follows.
>
> “We keep high-precision activation only on the skip connection and utilize it only for the element-wise addition. ” in Section 3.1.
> “In our proposed method, all of the element-wise multiplications in Equations 3e and 3f are performed in high precision.” in Section 3.2.
>
>
>
> 2. The novelty of the proposed method compared to Bi-Real Net
>
> Please refer to our response to reviewer 3.
>
>
>
> 3. Laplace distribution approximation
>
> Please note that the y-axis is in log-scale while x-axis in linear scale. The histogram decreases linearly in the plot, which is well modeled by Laplace distribution. The jitter at the ends is due to the fact that the number of samples is small, and the range is in log-scale. We performed the same quantization adopting other distributions including Gaussian and triangle distributions, and the Laplace distribution showed marginally better results than the others.
>
>
>
> 4. What kind of activation quantization is used?
>
> We use the conventional quantization method used in DoReFa-net, and the method is also adopted in Zhuang’s work. After clipping the activation to a pre-defined value, typically 1, the linear quantization is applied to the activation. We will clarify this in the revision.
>
>
>
> 5. when is the cosine similarity between the quantized and full-precision networks computed?
>
> Please refer to our response to reviewer 2.
>
>
>
> 6.  What are the axes in Figure 5? Why is there only one local minimum in Figure 5(d)? Why the training with PH converges even slower than without PH at the early stage of training?
>
> We appreciate the comments. It helped clarify the loss surface analysis in Figure 5. In order to obtain Figure 5, we applied Hao Li’s method as mentioned in the paper. In short, each figure represents loss surface seen from the local minimum we obtained from the training, i.e., the weight vector of the final trained model. In order to obtain two-dimensional view, we utilize two base vectors, u1 and u2, each of which corresponds to the axis of the figure. The base vector is a randomly generated vector having the same dimension of the weight vector. According to (Li et al., 2017), two randomly generated high-dimensional vectors tend to be orthogonal to each other. The origin of the figure at (0, 0) corresponds to the weight vector of the local minimum. The z-axis corresponds to the loss. In order to obtain a point, e.g., (0.25, 0.5) in the figure, we scale the two base vectors, i.e., 0.25*u1 and 0.5*u2, and add them to the local minimum weight vector corresponding to the origin. Then, we obtain the loss for the new weight vector, which is depicted at the point, (0.25, 0.5) on Figure 5. Since the figure is a loss surface near the local minimum, we tend to have a single local minimum in the figure unless we have another local minimum near the obtained one. The figure does not represent the relationship between loss and training epochs. We will clarify how we obtained Figure 5 in the revision.

---

### Official Review · AnonReviewer2 · 2018-11-05
**A paper with good ideas and solid results, but some overlap with the literature**

**Rating:** 7
**Confidence:** 4

**Review:**

This paper studies methods to improve the performance of quantized neural networks.  The paper is largely centered around the idea of "precision highways" (full-precision residual connections) that run in parallel to fully-quantized convolutions.  However, the paper also throws in a toolbox of other methods like distillation from a teacher network, a quantization method based on the Laplace distribution, and a fine tuning scheme.

The paper reports performance for the resulting networks that is impressive but still believable.   They also do very extensive experiments, including an ablation study in Table 1 that I really liked, and a study of how the precision of the skip connections impacts overall performance.   I also like the visualizations of how quantization impacts the loss surface.

My main concern about this paper is that is has conceptual overlap with other approaches.  The authors are not the first to quantize resnets, and other papers have looked at teacher training and distillation as a method of refinement.  The authors are fairly upfront about this though, and I think this paper is the first to do a really thorough investigation of the impacts of skip connections in their own right.    Realistically, fully binarizing neural nets without modification is unlikely to lead to good performance.  The idea of leaving the skip connections with higher precision is a good compromise that achieves hardware friendliness along with strong performance, so I think it's worth having a paper like this that takes a closer look at this approach.

A few questions I had:
1)  I can't tell exactly what methods are being used in Table 1.  When the "highway" box is unchecked, does this mean the skip connection is absent?  Or that it exists but with full precision?  Or maybe that the skip connection branched after the quantization instead of before?   Also, what fine-tuning methods is used when the "teacher" box is un-checked?

2) You implemented your own version of Zhuang's method.  However, I'd like to know how your numbers compare to the original reported numbers in Zhuang's paper.

One other minor criticism - When you fine-tune a modified network, the activations and weights will change.  It could be that the networks is modifying its parameters to account for (i.e., cancel out) the quantization errors.  For this reason I don't interpret Figure 4 as evidence for accumulation of error.  Perhaps this type of behavior would exists if you fine-tuned two full-precision networks using different random seeds, or different teacher networks.

---

> ### Author Response · Authors · 2018-11-19
> **The response for Reviewer 2**
>
>
> 1. The answer to question 1
>
> When the “highway” box is unchecked, the skip connection is branched after the quantization. This case corresponds to the conventional quantization where the quantization is combined with ReLU and, thus, skip connection is quantized before the branch. When the “teacher” box is unchecked, we use the conventional cross-entropy loss for training. We will clarify this in the revision.
>
>
>
> 2. The answer to question 2
>
> We re-implemented Zhuang’s baseline, but final accuracy is different due to the minor difference of implementation details on input augmentation and teacher-student methods. According to Zhuang’s paper, their implementation shows 70.8 /88.3 % of Top-1/Top-5 accuracy for 2-bit ResNet-50, while our implementation shows 70.48 / 89.93 % of Top1-/Top-5 accuracy.
>
>
>
> 3. The comments about Figure 4
>
> We appreciate the comments. We’d like to first explain how we had obtained Figure 4 and how we performed again new experiments to clarify the phenomenon of accumulated quantization error in the revision.
> In order to obtain Figure 4, we first obtained a fully trained full-precision network. Then, we applied 4-bit weight/activation quantization to the network while having two cases of skip connection, 4-bit one (Zhuang’s in the figure) and 32-bit one (Proposed in the figure). Since they are from the same fully trained full-precision network, we think that the difference between the two graphs in Figure 4 represents the effect of high-precision skip connection.
>
> In order to account for the reviewer’s comments and give a more direct comparison, we did new experiments where we prepared, from the same initial condition, two activation-quantized networks (one with precision highway and the other with low precision skip connection) where weights are not modified and only activations are quantized to 4 bits. The new experiments give a similar result to Figure 4 while the difference in accumulated quantization errors gets slightly reduced, possibly, due to the removal of the quantization error of weights.  In order to clarify the phenomenon of accumulated quantization error, we will use the new experimental results in the revision.

---

### Official Review · AnonReviewer3 · 2018-11-08
**An OK paper but need more evaluation.**

**Rating:** 6
**Confidence:** 5

**Review:**

This paper investigates the problem of neural network quantization. The main idea is to employ an end-to-end precision highway to reduce the accumulated quantization error and meanwhile enable ultra-low precision in deep neural networks.  The experimental results on the 3- and 2-bit quantizations of ResNet-18/50 and 2-bit quantization of an LSTM model demonstrate the effectiveness of the proposed method.

This paper is well written and organized. The idea of utilizing a high-precision information flow to reduce the accumulated quantization error is technically sound. The empirical studies on accumulated quantization error, loss surface analysis, model performance, and hardware cost are quite thorough and solid.

The idea of precision highway, however, is quite similar to the skip connections used in Bi-Real Net. Therefore, it may be a good idea to provide a thorough discussion over these two different methods so as to make the distinction.

In Table 2, the results of Bi-Real Net is based upon 1 bit activation/weight quantization, while the proposed method uses 2 bit activation/weight quantization. To give a fair comparison, it may be better to provide 1 bit activation/weight quantization results of the proposed method.

---

> ### Author Response · Authors · 2018-11-19
> **The response for Reviewer 3**
>
>
> 1. The difference between the proposed method and Bi-Real Net
>
> We’d like to let you know that this study was conducted in parallel with Bi-Real Net. This work was submitted to another conference and re-submitted to ICLR2019 after adding additional extensive experiments including loss surface analysis and hardware cost estimation. We respect the research outcome of Bi-Real Net and refer to it in the original manuscript.
>
> As mentioned in the original manuscript, both Bi-Real Net and PACTv2 apply quantization on pre-activation style residual net, the basic module of which is composed of batch-normalization (BN) – ReLU – convolution. Meanwhile, the end-to-end precision highway is a generalized “network-level structural” method applicable to not only pre-activation style network but also post-activation style network, having the conv-BN-ReLU module as stated in the original manuscript as follows.
>
> “in the case of feed-forward networks with identity path, our precision highway idea is applicable regardless of pre-activation or post-activation structure”
>
> In addition, it is a general method also applicable to recurrent network including LSTM and GRU. We described how the precision highway can be applied to LSTM in Section 3.2 and showed it significantly outperforms the existing quantization method on a language model.
> Since it is a novel structural method, it raises new challenges for further improvements as mentioned in Section 3.3 of the original manuscript as follows.
>
> “In the case of networks with multiple candidates for the precision highway, e.g., DenseNet, which has multiple parallel skip connections (Huang et al., 2017), we need to address a new problem of selecting skip connections to form a precision highway, which is left for future work.”
>
> We think the precision highway opened a new space of mixed precision neural network design where the precision of data representation, previously ignored, can now be jointly optimized with that of computation for further improvements of quantized networks.
>
>
>
> 2. 1-bit quantization result
>
> We performed 1-bit activation/weight quantization for the post-act style ResNet-18. For a fair comparison, we didn’t apply the teacher-student and progressive quantization method and instead adopted BN-retraining proposed in Bi-Real Net. Our 1-bit activation/weight ResNet-18 gives 56.73 / 80.11 % of Top-1/Top-5 accuracy, which is by 0.33 / 0.61 % higher than the result of Bi-Real Net, respectively. According to our observation, the final accuracy is degraded when adopting a tight approximation of the derivative of the non-differentiable sign function proposed by Bi-Real Net. Instead, the conventional quantization method proposed by DoReFa-net can improve results. This difference seems to result from the difference of the network structure and activation quantization function. We will add this result and analysis to the revision.

---

### Public Comment · (anonymous) · 2018-11-28
**Serious concerns about novelty of this paper**

This paper has serious concerns about novelty in its contributions. Although the authors already acknowledged that “precision highway” is not a new idea, they claimed that their main contributions – generalization of the idea to post-activation Resnet and LSTM as well as the in-depth analysis – are significant. However, their generalization is almost trivial or already done before, and their analysis is misleading or contradicting to the previous well-established analysis.

1) Lack of novelty in the claims about generalization
1.a) The authors claim that applying “precision highway” for RNNs is their novel contribution. But what it does is simply quantizing only inputs of matrix multiplication and not quantizing inputs to the cell-state (ct) in LSTM, which is the de-facto standard quantization scheme for LSTM inference. To name a few, [Hubara et al., 2016], [He et al., 2016b]* and [Kapur et al., 2017] quantize only input (weight and activation) of matrix multiplication, and [Shin et al., 2016] and [Lee et al., 2016] even have some preliminary study/observation on the sensitivity of cell-state to the quantization.
* Note that [He et al., 2016b] is already cited in this paper for accuracy comparison. This comparison is very misleading; the authors claim that their work achieves “much better” accuracies than [He et al., 2016b] since “precision highway provides more gain for a more aggressive quantization”. But it is clearly specified in [He et al., 2016] that only inputs to the matrix multiplication (i.e., W, ht, xt) are quantized (i.e., it also uses “precision highway”!).
[Hubara et al., 2016]: https://arxiv.org/pdf/1609.07061.pdf
[He et al., 2016b]: https://arxiv.org/pdf/1611.10176.pdf
[Kapur et al., 2017]: https://arxiv.org/pdf/1710.07706.pdf
[Shin et al., 2016]: https://arxiv.org/pdf/1512.01322.pdf
[Lee et al., 2016]: https://arxiv.org/pdf/1610.00552.pdf

1.b) The authors also claim that the idea of using full-precision data for the skip-connections has been demonstrated only for “pre-activation” Resnet, thus extending it to “post-activation” is a significant progress. But there is little innovation added in this extension; the principle of not quantizing data used in the skip connection is the same for both “pre-“ and “post-“ activation, and there’s little additional care needed than following the same principle to identify the location of quantization.

2) Misleading or contradicting outcome from the quantitative analysis
2.a) Analysis of accumulated quantization error is based on the comparison of eq(1) and eq(2). But this is a mere observation that not quantizing skip-connection can remove one of the error terms, leading to smaller quantization error. In other words, this analysis is mainly focusing on the number of quantized elements. (If the number of error terms is what matters most, one can think of not quantizing other parts of the network, e.g., one of the residual passes, so that overall quantization error is further reduced.) Fig 4 shows a natural outcome; quantizing less number of data elements results in smaller quantization error. Unfortunately, this analysis does not shed light on understanding which of the error terms -- error in the residual-path (e_r) vs the skip-connection (e) -- influences more on the convergence of training.

2.b) The second analysis is based on the visualization of the loss function, but the interpretation of the plots is contradicting to the well-established analysis. The authors claim that “precision highway gives better loss surface” so that the proposed method would help SGD to “quickly converge” and offer “better accuracy”. However, in Fig 5 (d), “with PH” shows sharper loss curve than “w/o PH”, which (according to [Li et al., 2018]) would indicate sharper minima. There are many analysis papers such as [Keskar et al. 2017] that proposed a strong correlation between sharp minima and the generalization gap leading to accuracy degradation. Based on these papers, therefore, the plots can be interpreted as "with PH" might suffer larger generalization gap than "w/o PH" due to sharper minima, which is conflicting with the authors' interpretation.
[Li et al., 2018] http://arxiv.org/abs/1609.04836
[Keskar et al. 2017] http://arxiv.org/abs/1609.04836

Although there are several pros in this paper, such as clarification of the location of quantization for “post-activation” Resnets, and the ablation study on the impact of “highway precision” with normal and wide Resnets, the above concerns on the major contributions of this paper raise a question if it has sufficient novelty for ICLR publication.

---

> ### Author Response · Authors · 2018-11-29
> **The response for the anonymous reviewer**
>
>
> We appreciate the comments and, for each issue, present our response as follows
>
> 1.a)
> In the followings, we present our response to this issue for each related paper mentioned in the comments.
> According to the following excerpt from [He et al.], the quantization is applied to all the outputs of tanh and sigmoid activation functions including Ht. Only Ct is not quantized just in order to avoid clipping.
>
> “Different from GRU, Ct can not be easily quantized, since the value is unbounded by not using activation function like tanh and the sigmoid function. This difficulty comes from structure design and can not be alleviated without introducing extra facility to clip value ranges. But it can be noted that the computations involving Ct are all element-wise multiplications and additions, which may take much less time than computing matrix products. For this reason, we leave Ct to be in floating point form.”
>
> In case of GRU, all internal signals including Ct are quantized. Thus, we think the method in [He et al.] was designed without considering the effect of end-to-end high precision flow.
>
> Our work is conceptually different from [He et al.] in that ours shows that the end-to-end flow is critical in low precision and, thus, aims at keeping signals on the end-to-end flow in high precision while [He et al.] applies quantization to all the outputs of activation functions without considering the importance of end-to-end flow. In addition, we also present an improved weight quantization based on Laplace distribution model to offer further improvements.
>
>
> In [Kapur et al., 2017], the authors mention that they followed [He et al.] as shown in the following excerpt.
>  “We use a quantization method very similar to that proposed in [6]”
>
>
> In [Hubara et al., 2016], which is one of the first papers on binary quantization, the authors did not apply the notion of end-to-end high precision flow since they used AlexNet and GoogLeNet (without skip connection). Regarding the RNN, they quantized both activations and weights as shown in the following excerpt.
>
> “Here we report on the first attempt to quantize both weights and activations by trying to evaluate the accuracy of quantized recurrent models trained on the Penn Treebank dataset.”
>
> We can also guess that they did not apply the concept of end-to-end high precision flow to the RNN since their quantized RNNs suffer from significant accuracy loss.
>
>
> In [Shin et al., 2016], the authors clarified that they did not apply the notion of end-to-end high precision flow according to the following excerpt.
>
> “The output ranges of the sigmoid and the tanh are limited by 0 to 1 and -1 to 1, respectively. The quantization step size ∆ is determined by the quantization level M. For example, if the signal word-length is two bits (M is four), the quantization points are 0/3, 1/3, 2/3, and 3/3 for the sigmoid and -1/1, 0/1, and 1/1 for the tanh. However signals of linear units are not bounded and their quantization range should be determined empirically. In our phoneme recognition example, each component of the input data is normalized to have zero mean and a unit variance over the training set. The input range is chosen to be from -3 to 3. One hot encoding is used for the input linear units in the language model example.”
>
> They applied quantization to all the signals, even the inputs.
>
>
> In [Lee et al., 2016], the authors mentioned that all the weights and activations are quantized to 8 and 6 bits as shown in the following excerpt.
>
> “In our work, the weights and the internal signals are quantized to 6 and 8 bits, respectively.”
>
> Since they quantized all the internal signals of RNN, we think they did not apply the notion of end-to-end high precision flow.
>
> In summary, the papers mentioned in the comments did not apply the notion of end-to-end high precision flow in the quantization.
>
>
> 1.b)
> We think considering the notion of end-to-end high precision flow on both pre and post-activation makes a difference that ours can also be applied to other types of networks including LSTM and GRU as well as CNNs with skip connections. That is why we could extend our idea to LSTM networks as shown in the paper.
>
> We’d like to stress again that, compared with other works mentioned in the comments, an ‘explicit’ consideration of end-to-end high precision information flow is our key difference, which enables very low-precision quantization in RNNs (LSTMs and GRUs) as well as CNNs (with skip connections).

---

> > ### Author Response · Authors · 2018-11-29
> > **The response for the anonymous reviewer - 2**
> >
> >
> > 2.a)
> > We think there are two issues to address here, why to select e in the two error terms in Eq (1) and the convergence of training.
> >
> > Regarding the selection, we chose to remove the error term e in Eq (1), i.e., keep the skip connection in high precision in order to reduce the overhead of high precision. The computation overhead of keeping high precision on precision highway is small as mentioned in Section 3.2 as follows.
> >
> > “the overhead of this high-precision element-wise multiplications is negligible compared with the matrix multiplication”
> >
> >
> > We also reported the energy overhead of precision highway in the experiments of Section 5.5 as follows.
> >
> > “In the 2-bit case where the overhead of precision highway is the largest, the precision highway incurs only 3.9 % additional energy consumption due to the high-precision data while offering 4.1 % better accuracy than the case that precision highway is not adopted.”
> >
> >
> > Regarding the second issue of convergence in training, we have an evidence that precision highway gives better convergence (i.e., validation error vs epochs) during training. The following table shows the additional validation error of ResNet-50 (with respect to the full precision model) obtained during training for the 2-bit quantization. Note that, we reduce bit precision every 15 epochs by following gradual quantization [Zhuang et al.], and adopt knowledge distillation with a teacher of ResNet-101. Thus, the negative error, i.e., better accuracy than the full precision ResNet-50 model, is due to the teacher effect. As the table shows, the precision highway consistently outperforms No Highway during the entire training. More importantly, it offers better results as the training continues, e.g., 0.72% at 45 epochs to 1.86% at 150 epochs. We will include these results into the final version of the paper.
> >
> > # epochs           15       30       45        60       75      90     105    120      135    150
> > Highway         -0.61  -0.84   -0.74    -0.21   1.60   1.30   1.12   1.20     1.59   2.45
> > No highway    0.34   -0.32   -0.02     0.61   3.41   2.85   2.48   2.75     3.13   4.31
> >
> >
> > 2. b)
> > We think sharpness is a rather relative concept and the sharpness very near the local minimum is important. In the figure, we wanted to demonstrate that, at the local minimum, i.e., zero point in the figure, the loss surface of our method is much smoother than that of the existing method. We will clarify this in the final version of the paper.

---

> > ### Public Comment · (anonymous) · 2018-12-01
> > **Authors' answer to 1.a) seems to be wrong**
> >
> >
> > Note that in page 7 of [He et al., 2016]'s paper, it is clearly mentioned that only xt, W, ht (which will become an operand of matmul) are quantized, as highlighted in this capture (https://www.dropbox.com/s/20zr3wklqtrco18/tmp.png?dl=0).
> >
> > This way of quantization is identical to what is shown in Fig 2(b): ht and xt are stacked and quantized, then multiplied with quantized W. Thus this work already shows the impact of "end to end flow"
> >
> > (Rebuttals to the other answers will soon be followed.)

---

> > > ### Author Response · Authors · 2018-12-02
> > > **Clarification of the answer**
> > >
> > >
> > > We really appreciate the comments. It helped clarify our contribution with respect to [He, et al.]. In summary, our method is different from [He, et al.] by exploiting the precision highway in (1) fine-tuning for LSTM and (2) improving the cell structure for GRU as follows.
> > >
> > > In LSTM, both [He, et al.] and our methods have the same inter/intra-time step operation as mentioned in the new comments. However, We exploited the precision highway in training the quantized LSTM model as follows. First, we trained it in full precision. After then, we quantized the LSTM model incrementally in multiple steps. During one of the step, we quantized only the LSTM cells while keeping, in high precision, the rest of the network, and applied fine-tuning. We exploited high-precision h_t by not quantizing the output of LSTM cells which is the input of the decoder layer. It is different from [He, et al.] since the un-quantized h_t is provided to the decoder layer. This training pipeline with the high-precision h_t offers significant improvements in accuracy compared with the conventional quantization where high-precision h_t is not exploited. We will clarify this in the final version of the paper.
> > >
> > > Second, in GRU, our precision highway is structurally different from [He, et al.]. It is because h_t is quantized in [He, et al.] while our precision highway keeps it in high precision, which makes us form the end-to-end high-precision flow across time steps. We will prepare new experiments of quantizing GRU models to compare [He, et al.] and ours and present them in the final version of the paper.

---

### Public Comment · (anonymous) · 2018-12-04
**Rebuttal to the authors' responses**


Thanks to authors for the responses to my question. Here's my rebuttal to your answers.

- It seems to be clear that the concept of Precision Highway for LSTM (at least for the LSTM cells) is very similar to the conventional quantization approaches, if not identical. In fact, the first answer of the authors alludes to their confusion in understanding [He et al., 2016]; they said "the quantization is applied to all the outputs of tanh and sigmoid activation functions", which is wrong. The authors claim that there is a slight difference in the decoding layer, but it is not sure how important it is since it is not discussed at all in the submitted paper but the authors brought up only at the comment.

- The authors also claim Precision Highway for GRU as their novel contribution, but without thorough demonstration. To claim novelty, it should provide in-depth ablation study on it.

- Similarly, the authors claim that Precision Highway for "post-activation" ResNet is a novel contribution. But as the authors admitted, it is an addition to the existing concept in Liu et al. (2018) and Choi et al. (2018b) on "pre-actication ResNet (and thus the authors did not even demonstrate "pre-activation" in their paper). It would be more interesting to see the application of Precision Highway for drastically different network structures, such as ResNext or DenseNet, but the authors conveniently postpone it as a future work.

- The key claim of the authors about novelty is an "explicit" consideration of the end-to-end highway precision, which implies that this concept is already presented but they are revisiting explicitly. To differentiate itself enough, the readers would expect in-depth analysis of why Precision Highway is working, with theoretical proof or rigorous numerical analysis.

But the analysis presented in this paper is VERY naive. The authors explained that their choice of high-precision skip connection is to reduce overhead, not because of in-depth theoretical study. The overhead reduction is already the motivation of prior work, Liu et al. (2018) and Choi et al. (2018b). The authors don't show theoretically why high-precision in skip-connection is essential; even in their answer, they just showed a table of convergence curve, which is another empirical outcome of Precision Highway, not the reason of why this works.

The authors also responded back the question about the loss surface, but their evaluation is very subjective. They said "the surface of our method is much smoother", but how much smooth is good? Does this guarantee any convergence property? The authors said, "the sharpness very near the local minimum is important". But why? (why sharpness very near the local minimum is more important than sharpness "not very near" the local minimum?)


- After all, this paper seems to be premature; 1) the main concept of Precision Highway is borrowed from prior works, 2) extension of the idea is very misleading (e.g., misunderstanding on prior LSTM quant) or weak (GRU or "pre-activation" ResNet without demonstration), and 3) the analysis is very naive (does not explain high precision in shortcut is essential). It is true that Precision Highway is an important topic, but the way this paper addresses it currently is quite concerning as the readers expect more in-depth research on it.

---

### Meta-Review · Area_Chair1 · 2018-12-16
**Area chair recommendation**

**Confidence:** 4
**Recommendation:** Reject

**Metareview:**

The submission proposes a strategy for quantization of neural networks with skip connections that quantizes only the convolution paths, while leaving the skip paths at full precision.  The approach can save computation through compressing the convolution kernels, while spending more on the skip connections.
Empirical results show improved performance at 2-bit quantization compared to a handful of competing methods.  Figure 5 provides some interpretation of why the method might be working in terms of "smoothness" of the loss surface (term not used in the traditional mathematical sense).

The paper seems to focus too much on selling the name "precision highway" rather than providing proper definitions of their strategy (a definition block would be a good first step), and there is little mathematical analysis of the consequences of the chosen approach.
There are concerns about the novelty of the method, specifically compared to Liu et al. (2018) and Choi et al. (2018b), which propose approximately the same strategy.  Footnote 1 claims that these works were conducted in parallel with the current submission, but it is unambiguously the case that Choi et al appeared on arXiv in May, and Liu et al. appeared in ECCV 2018 and on arXiv more than 30 days before the ICLR deadline, and can fairly be considered prior work https://iclr.cc/Conferences/2019/Reviewer_Guidelines

The reviewer scores were on aggregate borderline for the ICLR acceptance threshold.  On the balance, the paper seems to fall under the threshold due to insufficient novelty and analysis of the method.